# AVATAR CONCEPT SLIDER: CONTROLLABLE EDITING OF CONCEPTS IN 3D HUMAN AVATARS

## ABSTRACT

Text-based editing of 3D human avatars to precisely match user requirements is challenging due to the inherent ambiguity and limited expressiveness of natural language. To overcome this, we propose the Avatar Concept Slider (ACS), a 3D avatar editing method that allows precise editing of semantic concepts in human avatars towards a specified intermediate point between two extremes of concepts, akin to moving a knob along a slider track. To achieve this, our ACS has three designs: Firstly, a Concept Sliding Loss based on linear discriminant analysis to pinpoint the concept-specific axes for precise editing. Secondly, an Attribute Preserving Loss based on principal component analysis for improved preservation of avatar identity during editing. We further propose a 3D Gaussian Splatting primitive selection mechanism based on concept-sensitivity, which updates only the primitives that are the most sensitive to our target concept, to improve efficiency. Results demonstrate that our ACS enables controllable 3D avatar editing, without compromising the avatar quality or its identifying attributes.

## 1 INTRODUCTION

Creating high-fidelity human avatars and editing them according to user demands has shown its importance in multiple scenarios such as game development, film production and virtual character creation for the metaverse and live streaming applications Alam et al. (2023); Yan et al. (2023); Wan & Lu (2023). Due to its importance, there has been an increased demand for better control over the creation and editing of personalized digital avatars Kolotouros et al. (2023); Liu et al. (2023); Yang et al. (2024); Xu et al. (2024), e.g., changing the hair style, clothing, or adding some accessories to the avatar. To meet this demand, avatar editing has attracted much research attention Liao et al. (2024); Cao et al. (2023); Xu et al. (2024). Drawing inspiration from the successes in harnessing diffusion models for 3D content generation Poole et al. (2023); Wang et al. (2023), previous works Cao et al. (2023); Liao et al. (2024) often edit avatars by leveraging a text-to-image diffusion model and using score distillation sampling (SDS) optimization, achieving 3D avatar editing with only text inputs and a text-to-image diffusion model. Recent works further harness multimodal large language models to guide the SDS optimization Xu et al. (2024), or improve the SDS process Sunagad et al. (2024). Overall, these developments have led to advancements in text-based 3D avatar editing.

Despite the significant progress, existing 3D avatar editing methods Liao et al. (2024); Cao et al. (2023); Xu et al. (2024) are limited due to their reliance on text prompts as the sole guidance signals. Specifically, text prompts can be quite ambiguous, making it difficult to edit the avatar to *precisely align with* the users' expectations. For instance, to describe the desired hair length of a human avatar, users may prefer descriptive words such as 'long' or 'short' – which are ambiguous and limited in expressing the exact degree of length. Moreover, this difficulty in achieving precise editing with text alone becomes even more evident when manipulating complex concepts like 'sharpness' of facial features, or other adjectives which may be hard to put into words, e.g., 'kind' or 'evil'.

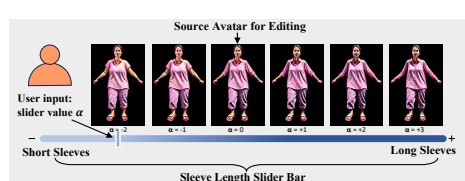

Figure 1: Illustration of our controllable concept editing. In this example, the user has provided text descriptions for two opposing ends of a concept: 'short sleeves' vs 'long sleeves'. Our ACS allows users to specify the exact level of concept expression (e.g., sleeve length) that is desired, by moving the knob on the slider bar.

Driven by the problems mentioned above, our aim is to *precisely edit target concepts* of avatars in a convenient manner, akin to moving the knob on a continuous slider bar, from which users may select *exact values along the bar* to specify their desired avatar. An overview is shown in Fig. 1. However, achieving this aim is challenging, because: ① our setting is challenging, where we aim to adjust only *a single parameter* for precise control during editing. Yet, for each concept (e.g., body shape), it can be difficult to find such a continuous bar to express the intermediate degree of avatar concepts, since these concepts are often high-level abstract concepts that exist beyond the pixel level. This makes it difficult to identify where these high-level concepts reside in feature space. ② Furthermore, it is challenging to isolate and edit only the desired concept without changing other identifying information of the avatar. This is because each avatar is generated with a mixture of elements, such as face shape, hairstyle, and clothing type, which are also entangled with concepts such as gender, race, and age. While editing the target concept, we may inevitably alter other attributes that we want to retain, and it is challenging to prevent this from happening.

In this work, we propose a novel Avatar Concept Slider (ACS) for 3D human full-body avatar editing, which enables users to edit their 3D avatar precisely towards the desired degree of expression of a given concept, offering much more controllable editing of concepts than with text inputs alone. To overcome the challenges highlighted above, our ACS comprises three corresponding designs, as follows: ① To pinpoint the precise concept axes that link the two opposing ends of a specific concept, we introduce the Concept Sliding Loss based on Linear Discriminant Analysis (LDA) to fine-tune an adapter. This facilitates precise slider-like editing using the SDS optimization editing pipeline. LDA facilitates the identification of continuous axes (which we call the concept axes) that are the most discriminative in linking the two opposing sides of the user-provided concept. ② To preserve key identifying attributes during editing, we further devise an Attribute Preserving Loss based on Principal Component Analysis (PCA), leveraging PCA to extract key attribute information from the features that are orthogonal to the target concept. Then, by encouraging these key attribute information to be retained, it enables users to edit their target concept in isolation, disentangled from other identifying attributes. ③ Moreover, we further propose a 3DGS primitive selection mechanism based on their concept-sensitivity, which enables us to selectively optimize a small set of the most crucial Gaussian primitives to reduce redundancy and improve efficiency.

With our method, users can fine-tune a concept slider by inputting descriptions of two opposing ends of a target concept, e.g., with a pair of text phrases. Then, using the trained slider, users can precisely adjust the degree of expression of a desired concept on a given 3D full-body avatar, by simply moving a knob across a slider bar, achieving controllable editing. We remark that, for each concept, the slider fine-tuning only needs to be done once, and can then be applied to any given 3DGS avatar.

## 2 RELATED WORK

**Text-Driven 3D Human Avatar Generation and Editing.** To streamline the labour-intensive 3D avatar creation process, many works Hong et al. (2022); Jiang et al. (2023); Zhang et al. (2023c;b); Kolotouros et al. (2023) focus on text-driven 3D avatar generation, relying solely on text prompt inputs to create a desired high-fidelity 3D human avatar. One line of works Hong et al. (2022) optimize their avatar via text guidance in CLIP space Radford et al. (2021). Subsequent works Kolotouros et al. (2023); Huang et al. (2023) rely on a text-to-image 2D diffusion prior (e.g., Rombach et al. (2022)) to optimize their 3D avatar via an SDS optimization pipeline. Notably, HumanGaussian Liu et al. (2023) proposes to use 3DGS to represent the human avatar, resulting in faster convergence and high-quality avatars. Recent works also adopt SDS optimization for editing. Some works focus specifically on the editing of human faces Mendiratta et al. (2023); Zhang et al. (2023a), sculpting faces according to the desired shape and texture. Several works also focus on editing full human avatars, through using SDS with guidance from a multimodal large language model Xu et al. (2024), with animatable human representations Sunagad et al. (2024), or by making the 3D avatar easily editable Yang et al. (2024); Liao et al. (2024). However, these text-driven avatar generation and editing methods are limited by the expressivity and ambiguity that is inherent in language, and often struggle with precisely generating the user-desired avatars with text prompts alone. Orthogonal to these works, our paper aims at improving the controllability of 3D avatar editing methods, enabling users to edit the avatar more precisely towards the desired degree of a target concept.

**Controllable Editing.** Controllable editing methods are increasingly crucial for enhancing generative flexibility in real-world applications. In the image domain, ControlNet Zhang et al. (2023d) is a

representative work which enables the flexible conditioning of image generation on different controls, beyond text-based inputs. Subsequent works Jiang et al. (2024); Zheng et al. (2023); Gandikota et al. (2024) explore this further, controlling the spatial layouts Zheng et al. (2023), injecting different conditions Jiang et al. (2024), or editing specific concepts in images Gandikota et al. (2024). In contrast, we are the first to explore controllable 3D human avatar editing, by precisely controlling their concepts via a slider. Our proposed ACS enables precise control via a slider, which is integrated into the SDS distillation pipeline, adjusting the concepts distilled into the 3D avatar.

## 3 BACKGROUND: SDS OPTIMIZATION PIPELINE

Score Distillation Sampling (SDS) Poole et al. (2023) leverages 2D diffusion models as a strong prior for 3D representation learning, e.g., text-to-3D content generation Lin et al. (2023) and 3D avatar editing Liao et al. (2024). Given a 3D representation $\theta$ (e.g., NeRF Mildenhall et al. (2020), 3DGS Kerbl et al. (2023)), SDS optimization pushes the renderings of the 3D representation towards the target distribution (e.g., high-quality natural images), by distilling the predicted score from a pre-trained 2D diffusion prior. Thus, by using text-to-image diffusion models Rombach et al. (2022), we can facilitate text-to-3D generation by optimizing the 3D representation via SDS.

Specifically, we consider the case where a latent diffusion model (e.g., Stable Diffusion Rombach et al. (2022)) is used, which is denoted as $\epsilon_\phi(\cdot)$. SDS optimization starts by rendering the initial 3D representation $\theta$ into 2D images, and encoding them into latent space to obtain the initial latent features as $\mathbf{z}_0$. Then, to derive the SDS loss $\mathcal{L}_{SDS}$, noise is added to the latent features $\mathbf{z}_0$ based on $t$ steps of forward diffusion Rombach et al. (2022) to obtain the noised latents $\mathbf{z}_t$, where the SDS loss aims to denoise $\mathbf{z}_t$. Therefore, the gradient of the SDS loss with respect to the 3D representation $\theta$ can be formulated as: $\nabla_\theta \mathcal{L}_{SDS}(\phi, \mathbf{z}_0) = \mathbb{E}_{t,\epsilon}[w(t)(\epsilon_\phi(\mathbf{z}_t; y, t) - \epsilon)\frac{\partial \mathbf{z}_0}{\partial \theta}]$, where $w(t)$ is a weighting function that depends on time step $t$, and $y$ is a text prompt that describes the desired 3D output.

Notably, to add functionalities, control options, or editing capabilities (e.g., with face landmarks Cao et al. (2023)) to the SDS optimization process while leveraging a pre-trained diffusion model, previous works use LoRA adapters Hu et al. (2022). Specifically, for a selected set of weight parameters (denoted as $\phi_0$) in the diffusion prior $\epsilon_\phi$, LoRA aims to learn a low-rank weight shift $\Delta\phi$ to obtain adapted weights $\hat{\phi}$ as: $\hat{\phi} = \phi_0 + \alpha\Delta\phi$, where $\alpha$ is the *scale factor* of the weight shift. By adapting the diffusion prior $\epsilon_\phi$ with $\Delta\phi$ and $\alpha$, the SDS optimization can be defined as:

$$\nabla_\theta \mathcal{L}_{SDS}(\phi, \mathbf{z}_0) = \mathbb{E}_{t,\epsilon}\left[ w(t)\Big(\epsilon_\phi\big(\mathbf{z}_t; y, t, (\Delta\phi, \alpha)\big) - \epsilon\Big) \frac{\partial \mathbf{z}_0}{\partial \theta} \right]. \tag{1}$$

Here, we also adopt the SDS optimization and LoRA adapter pipeline in Eq. 1.

## 4 METHOD: AVATAR CONCEPT SLIDER

Previous text-driven 3D avatar editing methods Liao et al. (2024); Xu et al. (2024) are limited in terms of precision of concepts. To overcome this limitation, this paper aims to achieve controllable manipulation of 3D avatars towards a target concept in a convenient manner, akin to sliding a knob along a slider bar. To address the challenges outlined in the introduction, we propose our Avatar Concept Slider (ACS), discussed in the following subsections. In Sec. 4.1, we address challenge ① by leveraging LDA and devising a sliding loss to enable slider-like precise control of the editing process. To tackle challenge ②, discussed in Sec. 4.2, we augment the sliding loss with an attribute-preserving loss based on PCA, ensuring concept-specific editing that is disentangled from the identifying attributes to be preserved. Lastly, in Sec. 4.3, we introduce a concept-sensitivity-based primitive selection mechanism, facilitating faster editing via selective optimization of our avatar's 3DGS primitives. Overall, our pipeline is built upon SDS optimization (introduced in Sec. 3), and our full method is illustrated in Fig. 2.

### 4.1 PRECISE SLIDER-LIKE CONCEPT EDITING

To precisely control the 3D avatar editing towards a specific degree of the target concept, the first challenge is to identify the continuous and discriminative axes connecting two opposing extremes of the concept. This can be challenging because the concepts are often abstract and high-level, making

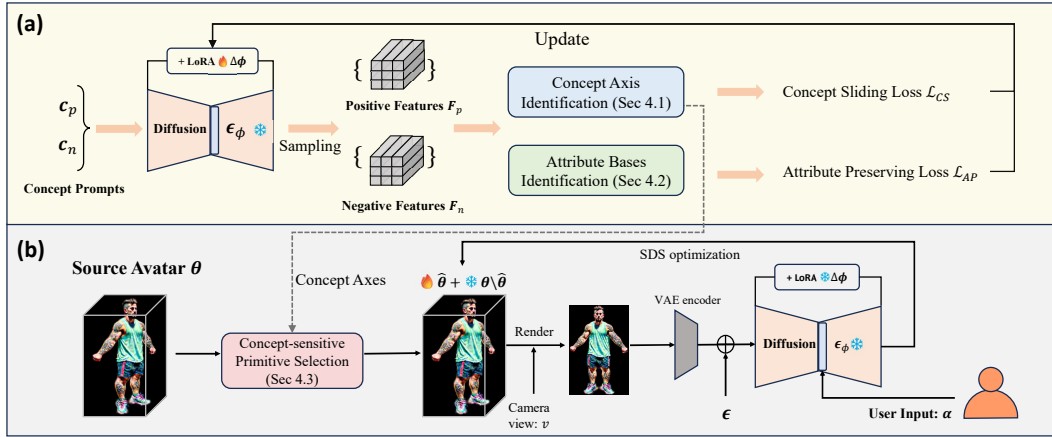

Figure 2: **(a)** Overview of the fine-tuning stage, where an adapter $\Delta\phi$ is fine-tuned to learn slider-like capabilities. Firstly, using the provided descriptions of the positive ($c_p$) and negative ($c_n$) side of the concept, we extract the corresponding positive and negative features. Then, during fine-tuning, our proposed Concept Sliding Loss (Sec. 4.1) helps learn the ability to slide across opposing ends of the target concept. The Attribute Preserving Loss (Sec. 4.2) helps to retain the avatar's key identifying attributes. **(b)** After fine-tuning, the adapter is applied during SDS optimization for controllable concept-specific 3D avatar editing. The proposed 3DGS primitive selection mechanism (Sec. 4.3) further improves efficiency, by optimizing only the most related primitives to the target concept.

it difficult to pinpoint where exactly these opposing sides and their connecting axes reside in the large feature space. To overcome this, we leverage LDA to identify the continuous concept axes between the user-provided pair of descriptions for the two opposing sides, by directly computing the axes that are the most discriminative to link the opposing sides within the large feature space. Then, based on the identified concept axes, we introduce a Concept Sliding Loss ($\mathcal{L}_{CS}$) to fine-tune an adapter, which enables controllable and precise concept-specific editing via SDS optimization (Sec. 3). Below, we first describe how we identify the concept axes, before discussing the Concept Sliding Loss.

**Concept Axes Identification with LDA.** First, we aim to identify the continuous concept axes between the user-provided descriptions that represent two opposing sides of the target concept, where each point on the axis represents an intermediate degree of expression of the concept. By identifying the concept axes, we can *precisely control the transitioning of the concept* from one end to the other, and facilitate the learning of sliding capability via the sliding loss, enabling users to select any intermediate point along the concept axes for precise editing. Yet, this task is not straightforward. For instance, a simple solution may be to interpolate the semantic embeddings of two concept extremes, but such simple interpolation operations in a high-dimensional space often incurs high information redundancy and concept entanglement, incorporating many other unrelated concepts.

Hence, to handle this challenge, we leverage LDA. LDA McLachlan (2005) is a statistical technique that aims to identify the most discriminative axes that well-separate different groups/classes of data samples, even while using a low dimensional representation. By employing LDA, we can directly compute a set of low-rank axes that are the most discriminative and crucial to link the opposing concepts within the large feature space. Specifically, we harness LDA to exploit the latent space of the pre-trained text-to-image diffusion model used during SDS optimization (Sec. 3) to facilitate understanding of the target concept in a high-quality latent space.

Here, we first present the details of how we identify the concept axes. For simplicity, we explain it in the context of two-class LDA – where only positive and negative prompts are provided, although we can potentially use more classes (explained at the end of Sec. 4.1). In this case, at the beginning, the user provides a pair of text prompts representing the positive ($c_p$) and negative ($c_n$) sides of a concept. Then, based on these prompts, we sample two sets of latent features from the diffusion model $\epsilon_\phi$ by running the generation process $N_s$ times for each of the positive and negative sides, where we perform sampling over different diffusion timesteps using different seeds. This yields two sets of features $\{\mathbf{F}_{p,i,x}\}_{i\in[1,N_s],x\in[1,C]}$ and $\{\mathbf{F}_{n,i,x}\}_{i\in[1,N_s],x\in[1,C]}$ corresponding to the positive and negative sides, where $C$ refers to the channel dimensions of latent features. Each feature vector is $D$-dimensional, i.e., $\mathbf{F}_{p,i,x} \in \mathbb{R}^D$, $\mathbf{F}_{n,i,x} \in \mathbb{R}^D$.

We then start the LDA analysis using these two feature sets for the two opposing sides of the target concept. Specifically, we first compute both the within-class scatter ($\mathbf{S}_w \in \mathbb{R}^{D \times D}$) and the between-class scatter ($\mathbf{S}_b \in \mathbb{R}^{D \times D}$), where $\mathbf{S}_w$ quantifies how much individual feature vectors deviate from their corresponding set centroids, while $\mathbf{S}_b$ measures the deviation between the centroids of both sides of the concept (more details in Supplementary). Then, using $\mathbf{S}_b$ and $\mathbf{S}_w$, the goal of LDA is to find *concept axis* $\mathbf{b}_c \in \mathbb{R}^D$, such that the projection of each feature vector onto the concept axis $\mathbf{b}_c$ leads to maximal ratio of $\mathbf{S}_b$ to $\mathbf{S}_w$. Intuitively, this encourages the feature vectors from both sets to be separated and positioned far from each other along the concept axis, while the feature vectors from the same set become tightly clustered along the concept axis. By doing so, LDA effectively pinpoints the direction of a concept axis, along which the opposing ends of the concept reside. Specifically, this process can be expressed as the following optimization problem:

$$\mathbf{b}_c = \arg\max_{\mathbf{w}} \frac{\mathbf{w}\mathbf{S}_b\mathbf{w}^T}{\mathbf{w}\mathbf{S}_w\mathbf{w}^T}, \text{s.t. } ||\mathbf{w}|| = 1, \tag{2}$$

where the numerator ($\mathbf{w}\mathbf{S}_b\mathbf{w}^T$) and denominator ($\mathbf{w}\mathbf{S}_w\mathbf{w}^T$) on the right hand side of Eq. 2 represent the projected scatters over the concept axis, and we want to find a concept axis $\mathbf{b}_c$ that maximises their ratio while constraining $\mathbf{w}$ to be a unit vector. Notably, the solution to Eq. 2 can be conveniently computed as the leading eigenvector of $\mathbf{S}_w^{-1}\mathbf{S}_b$ (refer to Supp).

In practice, we can use more than 2 classes/groups of features. For instance, we can include three classes: positive, neutral, and negative, allowing us to compute two concept axes via LDA that hold strong discriminative power between these classes, which are orthogonal to each other. We can then apply our sliding loss onto both concept axes for a stronger effect. Note that, even more classes can be included (with more prompts), but we found that using 3 classes is sufficient to obtain good results.

**Concept Sliding Loss.** Based on the concept axes identified in the above process, we introduce a Concept Sliding Loss to fine-tune an adapter to facilitate concept-specific editing of the 3D avatar. Since we adopt the SDS optimization pipeline (Sec. 3) to edit the 3D avatar via a pre-trained diffusion model, we aim to incorporate such concept-specific slider-like editing abilities into it. Hence, we propose a Concept Sliding Loss ($\mathcal{L}_{CS}$) to impart such concept transitioning abilities to the adapter, enabling us to precisely control the editing through a scale factor $\alpha$ in Eq. 1.

Intuitively, our Concept Sliding Loss $\mathcal{L}_{CS}$ facilitates the controlled movements of features along the concept axes towards a specified intermediate point, which is specified by the user as a *slider interpolation factor*. Specifically, to control such movements of features during editing, our loss fine-tunes the adapter ($\Delta\phi$) such that the scale factor $\alpha$ in Eq. 1 gains the ability as the *slider interpolation factor*. Thus, at deployment, with the user-provided interpolation factor $\hat{\alpha}$, we can set it as the scale factor (i.e., setting $\alpha = \hat{\alpha}$), thus allowing the adapted diffusion model to precisely locate the desired intermediate point on the concept axes. The Concept Sliding Loss $\mathcal{L}_{CS}$ can be defined as:

$$\mathcal{L}_{CS}(\Delta\phi) = \mathbb{E}_{\alpha_1, \alpha_2} \Big[ ||(\mathrm{Proj}(\mathbf{f}(\alpha_1, \Delta\phi, c_{neu}), \mathbf{b}_c) - \mathrm{Proj}(\mathbf{f}(\alpha_2, \Delta\phi, c_{neu}), \mathbf{b}_c))$$

$$-(\mathrm{Proj}\big(\frac{1+\alpha_1}{2}\boldsymbol{\mu}_p + \frac{1-\alpha_1}{2}\boldsymbol{\mu}_n, \mathbf{b}_c\big) - \mathrm{Proj}\big(\frac{1+\alpha_2}{2}\boldsymbol{\mu}_p + \frac{1-\alpha_2}{2}\boldsymbol{\mu}_n, \mathbf{b}_c\big))||^2 \Big], \tag{3}$$

where $\mathbf{f}(\cdot)$ represents the generation of latent features, $c_{neu}$ is a neutral text prompt template, $\boldsymbol{\mu}_p, \boldsymbol{\mu}_n \in \mathbb{R}^D$ are computed mean vectors for the positive and negative concept sides (see Supp for more details), and we select various points $\alpha_1, \alpha_2$ in the range $[-1, 1]$ to represent intermediate concepts; $\mathrm{Proj}(\cdot, \mathbf{b}_c)$ is the projection operator, which projects the features onto the concept axes $\mathbf{b}_c$, where $\mathrm{Proj}(x, \mathbf{b}_c) = \frac{x \cdot \mathbf{b}_c}{\mathbf{b}_c \cdot \mathbf{b}_c}\mathbf{b}_c^T$. Intuitively, this loss encourages the latent feature's movements along the concept axes (after applying the slider scale factor) to match the reference movements as calculated with the aid of the positive and negative concept text prompts. The latter is computed using the interpolations between $\boldsymbol{\mu}_n$ and $\boldsymbol{\mu}_p$ as: $\big(\frac{1+\alpha_1}{2}\boldsymbol{\mu}_p + \frac{1-\alpha_1}{2}\boldsymbol{\mu}_n\big)$ and $\big(\frac{1+\alpha_2}{2}\boldsymbol{\mu}_p + \frac{1-\alpha_2}{2}\boldsymbol{\mu}_n\big)$. Moreover, our loss is designed to encourage precise movements of the slider, and do not penalize the actual locations of latent features on the concept axes. This minimizes "calibration" issues during training where the source avatar is not neutral, e.g., when an "old man" is given the source avatar, we do not assume that the source avatar is at scale factor of "0".

Overall, this loss can fine-tune the adapter parameters $\Delta\phi$ such that the SDS optimization pipeline in Eq. 1 can be used to smoothly edit our avatar by adjusting the scale factor $\alpha$ during editing. Note that, we only need to identify the concept axes once (via LDA), and it can be used for the entire adapter fine-tuning process. Please refer to Supp for more details.

## 4.2 PRESERVING OF IDENTITY-RELATED ATTRIBUTES

Furthermore, to better maintain the identifying attributes of the avatar during editing, we propose an Attribute Preserving Loss. This loss is based on Principal Component Analysis (PCA), which helps disentangle the concepts of the avatar to facilitate retaining of key identifying attributes. Specifically, PCA helps to identify a small set of highly informative bases that are orthogonal to our concept axes, *representing key identifying attributes unrelated to the target concept*, but which can explain the other key variations in the latents. Based on these key attribute bases found via PCA, we propose an Attribute Preserving Loss $\mathcal{L}_{AP}$, enabling fine-tuning of the adapter to preserve identifying attributes. Below, we first discuss the bases identification with PCA, followed by the attribute preserving loss.

**Attribute Bases Identification with PCA.** Firstly, we reuse the features ($\mathbf{F}_{p,i,x}$ and $\mathbf{F}_{n,i,x}$) that were computed in Sec. 4.1, which represent the positive and negative sides of the target concept. We run PCA analysis on these features to find the $K$ principal components (i.e., orthogonal bases $\{\mathbf{b}_{a,k}\}_{k=1}^{K}$) which are orthogonal to our concept axes $\mathbf{b}_c$. Specifically, one way of performing PCA is through iteratively searching for the next principal component, which explains as much of the remaining variance of the data as possible, while being orthogonal to the previous identified principal components. Here, we recurrently employ this PCA technique to compute the main component bases orthogonal to our concept axes $\mathbf{b}_c$, formulated as:

$$\mathbf{b}_{a,k} = \arg\max_{\mathbf{w}} \frac{\mathbf{w}\hat{\mathbf{F}}_k\hat{\mathbf{F}}_k^T\mathbf{w}^T}{\mathbf{w}\mathbf{w}^T}, \text{ where } \hat{\mathbf{F}}_k = \hat{\mathbf{F}} - \sum_{j=1}^{k-1} \hat{\mathbf{F}}\mathbf{b}_{a,j-1}^T\mathbf{b}_{a,j-1}, \tag{4}$$

where $\hat{\mathbf{F}} \in \mathbb{R}^{D \times (2 \cdot N_S \cdot C)}$ is the combined merger of both positive and negative sets of features ($\{\mathbf{F}_{p,i,x}\}_{i\in[1,N_s],x\in[1,C]}$ and $\{\mathbf{F}_{n,i,x}\}_{i\in[1,N_s],x\in[1,C]}$), and $\mathbf{w}$ is constrained to be a unit vector. Crucially, by treating the first 'principal component' $\mathbf{b}_{a,0}$ as our concept axes $\mathbf{b}_c$ identified in Eq. 2 and iteratively solving Eq. 4 for $k = \{1, ..., K\}$, we can obtain $K$ key attribute bases $\{\mathbf{b}_{a,k}\}_{k=1}^{K}$ that best explain the remaining variance in the features. These obtained bases are also orthogonal to our concept axes $\mathbf{b}_c$ which are the key discriminative concept axes, which suggests that these bases are unrelated to the target concept.

**Attribute Preserving Loss.** With the identified attribute bases ($\{\mathbf{b}_{a,k}\}_{k=1}^{K}$) above that represent important key identifying attributes which should be retained, we introduce our attribute preserving loss to fine-tune the adapter ($\Delta\phi$) to disentangle its editing effects, avoiding the editing of identifying attributes of the avatar. In other words, when users adjust the scale factor ($\alpha$) to perform editing, we expect the other key attributes to remain unchanged, as if the adapter is not applied for those other attributes. Thus, the attribute preserving loss $\mathcal{L}_{AP}$ can be defined as:

$$\mathcal{L}_{AP}(\Delta\phi) = \mathbb{E}_{\alpha}\Big[ \sum_{k=1}^{K} ||\operatorname{Proj}(\mathbf{f}(\alpha, \Delta\phi, c_{neu}), \mathbf{b}_{a,k}) - \operatorname{Proj}(\mathbf{f}(\varnothing, c_{neu}), \mathbf{b}_{a,k})||^2 \Big], \tag{5}$$

where $\mathbf{f}(\varnothing, c_{neu})$ represents the original latent features without applying the adapter, and various values of $\alpha$ are selected in the range $[-1, 1]$. Intuitively, this loss compares the features after slider adaptation with the features that have not undergone adaptation, comparing them along a few key orthogonal dimensions that represent the key attributes of the avatar which should be retained. This encourages the adapter parameters ($\Delta\phi$) to retain main attributes unrelated to the concept axes.

Overall, by applying both the sliding loss $\mathcal{L}_{CS}$ (Eq. 3) and the attribute preserving loss $\mathcal{L}_{AP}$ (Eq. 5) to fine-tune the adapter $\Delta\phi$ (see Fig. 2(a)), the model gains the ability to perform precise concept-specifc editing via the SDS optimization pipeline in Eq. 1. Thus, when the user manipulates the scale factor $\alpha$ at deployment time, the model can edit the 3D avatar towards a precise point along the concept axis, while maintaining the key identifying features. We remark that, for each concept, the adapter fine-tuning only needs to be done *once*, and can then be applied to any given 3DGS avatar.

## 4.3 CONCEPT-SENSITIVE 3DGS PRIMITIVE SELECTION

Moreover, we explore a technique to reduce redundant computations during avatar editing, further improving the efficiency of 3D human avatar editing during SDS optimization, specifically for avatars in the 3D Gaussian Splatting (3DGS) representation.

3DGS is known for its efficiency in rendering and optimization Liu et al. (2023); Moreau et al. (2024), and has recently been of great interest for generation of 3D objects Tang et al. (2023);

Chen et al. (2024) and 3D humans Liu et al. (2023). Thus, considering its efficiency, we adopt the 3DGS representation for our 3D avatars. The 3DGS representation consists of a set of $M$ Gaussian primitives which we denote as $\{\boldsymbol{\theta}_i\}_{i=1}^M$, where each $i$-th Gaussian primitive contains a set of parameters $\{\boldsymbol{\mu}_i, \boldsymbol{\Sigma}_i, \sigma_i, \mathbf{h}_i\}$ that determine their location ($\boldsymbol{\mu}_i$), shape ($\boldsymbol{\Sigma}_i$), opacity ($\sigma_i$), and color ($\mathbf{h}_i$). During rendering, the color of each pixel is computed based on all the parameters of all $M$ primitives $\{\boldsymbol{\theta}_i\}_{i=1}^M$ in a differentiable manner, which facilitates the backpropagation of gradients.

Yet, we find that optimizing the entire 3DGS representation (i.e., all $M$ primitives) yields sub-optimal results for 3D editing. This is because, not all primitives contribute to the final rendering equally Niedermayr et al. (2024), some of them are not so important, and so optimizing all primitives incurs high redundancy and harms optimization and editing efficiency, In addition, our concept-specific 3D editing task typically needs to modify only a small subset of primitives that contribute the most towards the target concept. Thus, we select and edit only a subset of primitives, as discussed below.

**Concept-Sensitive Primitive Selection.** We propose a primitive selection mechanism based on concept-sensitivity to further improve the efficiency of the 3DGS representation in our editing pipeline. To achieve this, we leverage the concept axes identified in Sec. 4.1 to quantify the contribution for each primitive, by computing a *Concept Sensitivity score* that measures how much each primitive is related to the target concept, i.e., how the changes in the primitive parameters affect the precise location along the concept axis. Then, we select primitives with a high Concept Sensitivity score and edit only these primitives, effectively reducing the redundancy during the editing stage.

Firstly, to compute the sensitivity of Gaussian primitives with respect to the target concept, we need to express our 3D avatar in the same latent space as the identified concept axis $\mathbf{b}_c$. To do this, we first render our 3D avatar – which is represented by a set of primitives $\{\boldsymbol{\theta}_i\}_{i=1}^M$ – to $V$ randomly sampled views, yielding a set of rendered images. These rendered images are then passed into the diffusion model, to extract a set of latent features as $\{\mathbf{z}_{v,x,y}\}_{v\in[1,V], x\in[1,C]}$, where $\mathbf{z}_{v,x} \in \mathbb{R}^D$. These latent features $\mathbf{z}_{v,x}$ belong to the same latent space as $\mathbf{F}_{p,i,x}$ and $\mathbf{F}_{n,i,x}$ in Sec. 4.1.

Next, we calculate the Concept Sensitivity score based on these latent features. Intuitively, the Concept Sensitivity score measures the sensitivity of each primitive towards the target concept, i.e., how much movement there is along the identified concept axis $\mathbf{b}_c$ given a small change in the primitive's parameters. To compute this, we first compute the projection of the features $\mathbf{z}_{v,x}$ onto the concept bases $\mathbf{b}_c$ as follows: $\bar{C} = \frac{1}{V} \sum_{v=1}^V \mathrm{Proj}(\mathbf{z}_v, \mathbf{b}_c)$. The higher the values in $\bar{C}$, the more aligned the 3D human avatar is towards the target concept. Hence, if we take a gradient of $\bar{C}$ with respect to the primitive parameters, it will *measure how much movement* there is along the concept axes, given a small change in the primitive, i.e., the sensitivity of the target concept with respect to each primitive's parameters. Specifically, for each $i$-th Gaussian primitive $\boldsymbol{\theta}_i$ with parameters $\{\boldsymbol{\mu}_i, \boldsymbol{\Sigma}_i, \boldsymbol{\theta}_i, \mathbf{h}_i\}$, we compute its Concept Sensitivity score $S_i$ as:

$$S_i = \sum_{p_i \in \{\boldsymbol{\mu}_i, \boldsymbol{\Sigma}_i, \sigma_i, \mathbf{h}_i\}} \sum_{l=1}^L \left| \frac{\partial \bar{C}_l}{\partial p_i} \right|, \tag{6}$$

which sums the magnitude of the gradients w.r.t. each parameter, where $L$ is the dimensionality of $\bar{C}$. The higher the Concept Sensitivity score $S_i$, the more movement there will be along the concept axes given a small change in the $i$-th Gaussian primitive, which means that primitive is more crucial to editing of the concept. To select the most important primitives to edit, we pre-define a small fractional threshold $\gamma$, such that only the top $\gamma M$ primitives, i.e., $\{\boldsymbol{\theta}_i\}_{i=1}^{\gamma M}$, are selected for SDS optimization, instead of all $M$ primitives. Overall, this leads to significantly improved efficiency.

## 5 IMPLEMENTATION DETAILS

**Fine-tuning Details.** We use the Depth-RGB dual branch diffusion model Liu et al. (2023) as our diffusion prior. During fine-tuning, we weight the sliding loss and attribute preserving loss equally (coefficient of $0.5$ for each), and use LoRA adapter of rank 8. To maintain good avatar quality, we also include a quality preservation loss, which encourages the output of the slider at value 1 to match the output of the base diffusion model with positive prompts, maintaining the generation quality. We set $N_s = 40$, and sample positive, negative and neutral prompts. We set $K = 5$. The adapter is fine-tuned for 1000 steps with batch size 1, using AdamW optimizer (learning rate $= 2e - 4$). For *each concept-specific adapter*, the whole fine-tuning process takes 1 hour and is *executed once only*.

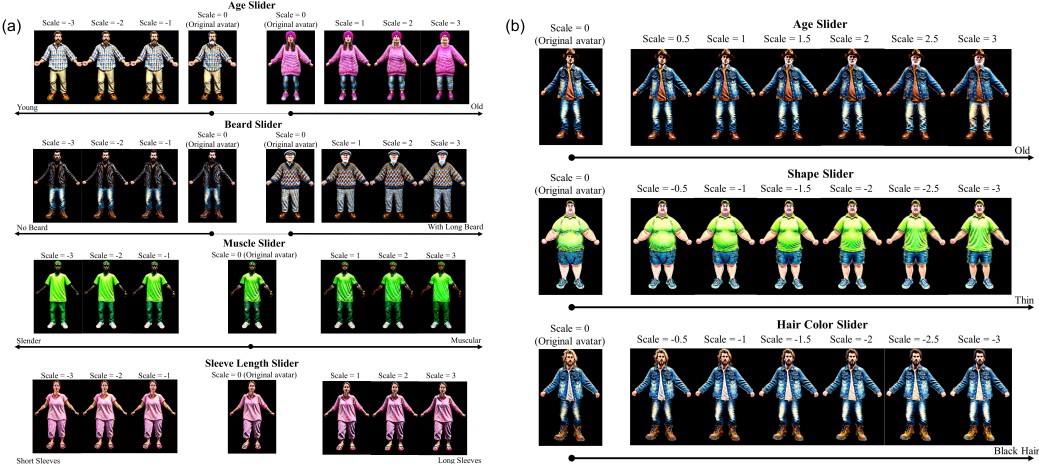

Figure 3: Editing results with concept sliders: (a) various concepts and (b) precise controllability.

**Avatar Editing Details.** We base our 3DGS implementation on the ThreeStudio framework Guo et al. (2023) and the optimized renderer from Kerbl et al. (2023). To obtain source avatars to edit, we generate 3DGS avatars with Liu et al. (2023). Then, following the optimization settings of Liu et al. (2023), we sample the camera distance from range $[1.5, 2.0]$, elevation from range $[40, 70]$, and azimuth from range $[-180, 180]$ with batch size of 4, for each editing update. We set $\gamma = 0.2$. The whole editing process undergoes 1200 updating steps (of SDS optimization). The whole editing process costs 12 minutes on a single NVIDIA 4090 GPU. Please see Supplementary for more details.

## 6 EXPERIMENT RESULTS

Here, we report and discuss our main results. Please *refer to Supplementary for more comprehensive results*, and *videos on our anonymized webpage*: *https://avatarconceptslider.github.io/*.

**Qualitative Results.** To evaluate our method, we first present editing results across various concepts and avatars in Fig. 3(a). For each concept, we show results for editing a given avatar by sliding at equally spaced intervals towards the positive (+1, +2, +3) and negative (-1, -2, -3) sides. Results show that our proposed ACS is capable of producing precise changes within the provided avatars effectively, while maintaining their quality and identity. Also note that we only fine-tune our LoRA slider within the range [-1,1], yet our ACS shows strong extrapolation ability beyond this range, indicating a well-generalized understanding of the concept.

**Precise Editing Results.** Next, we further verify our ability for precise control over concepts in Fig. 3(b), where we adjust the slider in smaller step sizes of 0.5. Our ACS is capable of producing minor changes given small increments in the scale, showing its efficacy at editing concepts precisely.

**Comparisons with Baseline.** Here, we run baseline comparisons against text-based editing baselines: HumanGaussian Liu et al. (2023), DreamAvatar Cao et al. (2023), TADA Liao et al. (2024). We adopt their methods while altering the text prompts to perform text-based editing, where we modify the text prompts by adding concept-related words (e.g., young, old) and adding adverbs to express degrees of expression, e.g., 'very', 'extremely'. Source 3DGS avatars for our method are generated using the pipeline of Liu et al. (2023). Source avatars for each baseline method are obtained using their own generation pipeline, such that the avatars are in the correct representation, and we do not introduce errors through conversion; text prompts used for the source avatars in direct comparison are the same. Our comparisons against the baselines are presented in Fig. 4(a). Crucially, unlike the text-driven baselines, our method offers *precise controllability*, enabling users to select a slider factor to edit the avatar concept to a desired degree. Our edited outcomes also tend to *better align with the provided descriptions*. Please refer to Supplementary for more comparisons.

**User Study.** We conduct a user study to show the superiority of our ACS for manipulating the degree of concepts. We invited 30 individuals from the general public to compare our method with a

Table 1: Results of user study.

| Setting | User Preference Rate (%) | | |
|---|---|---|---|
| | Q1 | Q2 | Q3 |
| Baseline Liu et al. (2023) | 15.62 | 16.67 | 26.04 |
| Ours | **84.38** | **83.33** | **73.96** |

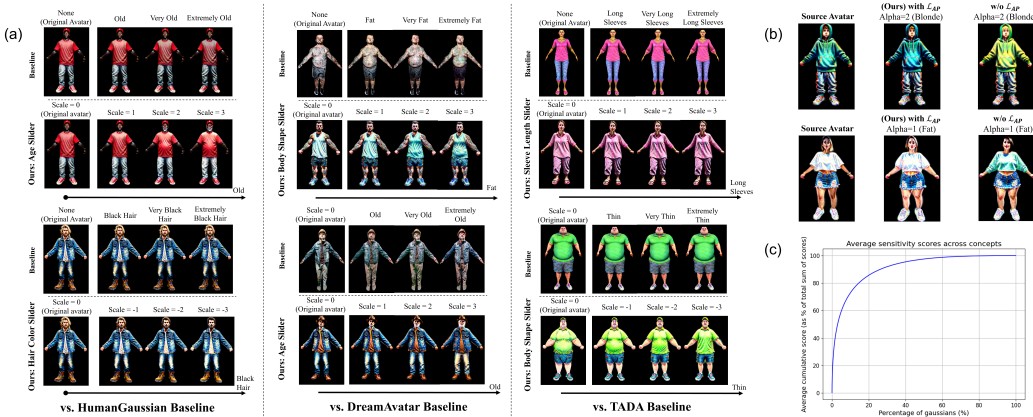

Figure 4: (a) Comparison of our method against existing text-based baselines Liu et al. (2023), Cao et al. (2023), Liao et al. (2024). (b) Impact of Attribute Preserving Loss $\mathcal{L}_{AP}$. (c) Average percentage of Gaussian primitives needed to account for sensitivity scores.

text-based baseline Liu et al. (2023). Each participant was shown 16 sets of edited avatars, with each set containing results from both methods edited to various degrees of a concept. Participants answered three questions regarding editing relevance and quality: (1) Which image shows more relevant results to the editing target? (2) Which image better shows the concept changing from source to target? (3) Which image shows better editing quality? As shown in Tab. 1, users significantly favour our ACS over the baseline.

## 7 MORE EXPERIMENTS AND ABLATION STUDIES

Below, we present more experiment results and ablation studies. Please *refer to Supplementary for even more studies and results*.

**Impact of Attribute Preserving Loss.** In Sec. 4.2, we proposed the Attribute Preserving Loss $\mathcal{L}_{\mathcal{AP}}$ to better retain identity information during editing. We visualize the effect of $\mathcal{L}_{\mathcal{AP}}$ in Fig. 4(b), where we observe that it helps to better retain identity information (e.g., clothing color) during the editing.

**Analysis of Concept-Sensitive Primitive Selection.** In our ACS, we further introduce the concept-sensitive selection mechanism to edit only a small fraction of primitives based on the Concept Sensitivity score. To find out how many primitives are sensitive to the concept on average, we computed the Concept Sensitivity scores of primitives across 5 concepts,

Table 2: Efficiency comparisons of our method.

| Setting | Time Taken to Converge |
|---|---|
| HumanGaussian Liu et al. (2023) | 42 min |
| AvatarVerse Zhang et al. (2023b) | 91 min |
| DreamAvatar Cao et al. (2023) | 4 hr |
| TADA Liao et al. (2024) | 2 hr |
| Ours | 12 min |

where 10 avatars were used for each concept. We summed up the Concept Sensitivity scores (which are non-negative) for each avatar and concept, and plotted the average percentage of primitives required to get a certain percentage of the summed sensitivity scores in Fig. 4(c). We observe that, on average, across concepts and avatars, about 20% of the primitives account for 85% of the summed Concept Sensitivity scores, which means they receive 85% of the total gradients. Therefore, we set the fractional threshold $\gamma$ to 0.2 to focus on editing these "concept-sensitive" primitives, improving efficiency while maintaining editing quality. Moreover, Tab. 2 compares the editing speed of our method against other methods for text-based editing, including methods using implicit representations Zhang et al. (2023b); Cao et al. (2023), parametric models Liao et al. (2024), and 3DGS Liu et al. (2023). Results show that our ACS is significantly more efficient than the baselines.

## 8 CONCLUSION

In this work, we introduce our novel ACS for 3D avatar editing, enabling users to edit 3D full-body avatars precisely towards the desired degree of expression of a given concept. Our ACS includes a Concept Sliding Loss based on LDA and Attribute Preserving Loss based on PCA. We also introduce a concept-sensitivity selection mechanism for improved editing efficiency. Results show that ACS enables controllable and concept-specific avatar editing, while maintaining the quality of the avatars.

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
