# AVATAR CONCEPT SLIDER: CONTROLLABLE EDITING OF CONCEPTS IN 3D HUMAN AVATARS

## 1 MORE DETAILS ABOUT LDA

In Sec. 4.1 of the main paper, we present how we leverage Linear Discriminant Analysis (LDA) to identify a concept axis to link the opposite sides of a concept. Here, we discuss more regarding the details of LDA.

First, we provide more details regarding the within-class scatter ($\mathbf{S}_w$) and the between-class scatter ($\mathbf{S}_b$) which are used in Eq. 2. As discussed in the main paper, the within-class scatter ($\mathbf{S}_w$) quantifies how much individual feature vectors deviate from their corresponding set centroids, while the between-class scatter ($\mathbf{S}_b$) measures the deviation between the centroids of both sides of the concept.

More precisely, given the two sets of features $\{\mathbf{F}_{p,i,x}\}_{i\in[1,N_s],x\in[1,C]}$ and $\{\mathbf{F}_{n,i,x}\}_{i\in[1,N_s],x\in[1,C]}$, the within-class scatter $\mathbf{S}_w$ is calculated as the summation of the variance for each set as:

$$
\begin{aligned}
\mathbf{S}_w &= \sum_{i=1}^{N_s}\sum_{x=1}^{C}(\mathbf{F}_{p,i,x}-\boldsymbol{\mu}_p)(\mathbf{F}_{p,i,x}-\boldsymbol{\mu}_p)^T \\
&+ \mathbf{S}_w = \sum_{i=1}^{N_s}\sum_{x=1}^{C}(\mathbf{F}_{n,i,x}-\boldsymbol{\mu}_n)(\mathbf{F}_{n,i,x}-\boldsymbol{\mu}_n)^T,
\end{aligned}
\tag{1}
$$

where $\boldsymbol{\mu}_p, \boldsymbol{\mu}_n \in \mathbb{R}^D$ are computed mean vectors for each concept side, i.e., $\boldsymbol{\mu}_p = \frac{1}{N_s C}\sum_{i=1}^{N_s}\sum_{x=1}^{C}\mathbf{F}_{p,i,x}$ and $\boldsymbol{\mu}_n = \frac{1}{N_s H C}\sum_{i=1}^{N_s}\sum_{x=1}^{C}\mathbf{F}_{n,i,x}$. At the same time, we can calculate the between-class scatter $S_b$ as:

$$
\mathbf{S}_b = (\boldsymbol{\mu}_p-\boldsymbol{\mu}_n)^T(\boldsymbol{\mu}_p-\boldsymbol{\mu}_n)\text{where } \bar{\boldsymbol{\mu}} = (\boldsymbol{\mu}_p+\boldsymbol{\mu}_n)/2
\tag{2}
$$

Then, based on the computed $\mathbf{S}_w$ and $\mathbf{S}_b$, we seek to identify the concept basis $\mathbf{b}_c$, such that the projection of each feature vector onto the concept axis $\mathbf{b}_c$ leads to maximal ratio of $\mathbf{S}_b$ to $\mathbf{S}_w$. Therefore, as mentioned in Eq. 2 of main paper, this process can be expressed as the following optimization problem:

$$
\mathbf{b}_c = \arg\max_{\mathbf{w}} \frac{\mathbf{w}\mathbf{S}_b\mathbf{w}^T}{\mathbf{w}\mathbf{S}_w\mathbf{w}^T}, \text{s.t. } ||\mathbf{w}|| = 1,
\tag{3}
$$

Notably, it can be shown that the solution to Eq. 3 can be conveniently computed as the leading eigenvector of $\mathbf{S}_w^{-1}\mathbf{S}_b$, which we provide the derivation below.

First of all, we define $\mathcal{J}(\mathbf{w}) = \frac{\mathbf{w}\mathbf{S}_b\mathbf{w}^T}{\mathbf{w}\mathbf{S}_w\mathbf{w}^T}$. Then, we take the derivative of $\mathcal{J}(\mathbf{w})$ with respect to $\mathbf{w}$ and set it to zero, so that we can have:

$$
\begin{aligned}
\frac{d\mathcal{J}(\mathbf{w})}{d\mathbf{w}} &= \frac{(\frac{d}{d\mathbf{w}}\mathbf{w}\mathbf{S}_b\mathbf{w}^T)\mathbf{w}\mathbf{S}_w\mathbf{w}^T - (\frac{d}{d\mathbf{w}}\mathbf{w}\mathbf{S}_w\mathbf{w}^T)\mathbf{w}\mathbf{S}_b\mathbf{w}^T}{(\mathbf{w}\mathbf{S}_w\mathbf{w}^T)^2} \\
&= \frac{(2\mathbf{S}_b\mathbf{w}^T)\mathbf{w}\mathbf{S}_w\mathbf{w}^T - (2\mathbf{S}_w\mathbf{w}^T)\mathbf{w}\mathbf{S}_b\mathbf{w}^T}{(\mathbf{w}\mathbf{S}_w\mathbf{w}^T)^2} = 0
\end{aligned}
\tag{4}
$$

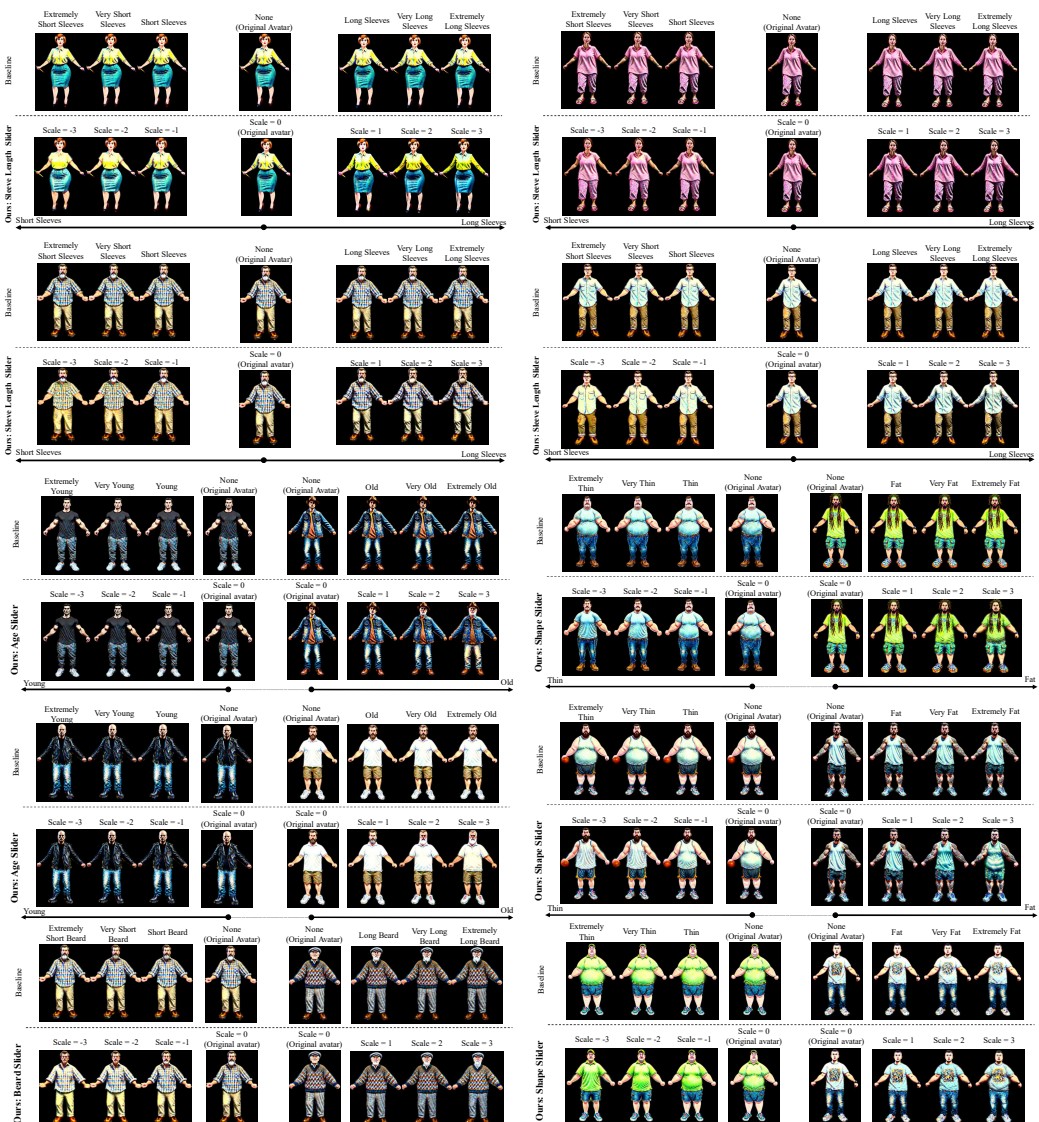

Figure 1: More qualitative comparisons with text-based baseline Liu et al. (2023).

Then, we simplify it further as follows:

$$(2\mathbf{S}_b\mathbf{w}^T)\mathbf{w}\mathbf{S}_w\mathbf{w}^T - (2\mathbf{S}_w\mathbf{w}^T)\mathbf{w}\mathbf{S}_b\mathbf{w}^T = 0$$

$$\mathbf{S}_b\mathbf{w}^T - \frac{\mathbf{w}\mathbf{S}_b\mathbf{w}^T(\mathbf{S}_w\mathbf{w}^T)}{\mathbf{w}\mathbf{S}_w\mathbf{w}^T} = 0 \tag{5}$$

$$\mathbf{S}_w^{-1}\mathbf{S}_b\mathbf{w}^T = \mathcal{J}\mathbf{w}^T$$

Therefore, by applying the definition of eigenvectors, we find that the stationary points of $\mathcal{J}(\mathbf{w})$ only occur when $\mathbf{w}$ is an eigenvector of $\mathbf{S}_w^{-1}\mathbf{S}_b$. In other words, over all the stationary points, $\mathcal{J}(\mathbf{w})$ takes values corresponding to all the eigenvalues of $\mathbf{S}_w^{-1}\mathbf{S}_b$. As a result, the maximum value of $\mathcal{J}(\mathbf{w})$ is equivalent to the largest eigenvector, and happens when $\mathbf{w}$ is the leading eigenvector of $\mathbf{S}_w^{-1}\mathbf{S}_b$. Hence, this shows that Eq. 3 can be solved by simply setting $\mathbf{w}$ to be the leading eigenvector of $\mathbf{S}_w^{-1}\mathbf{S}_b$.

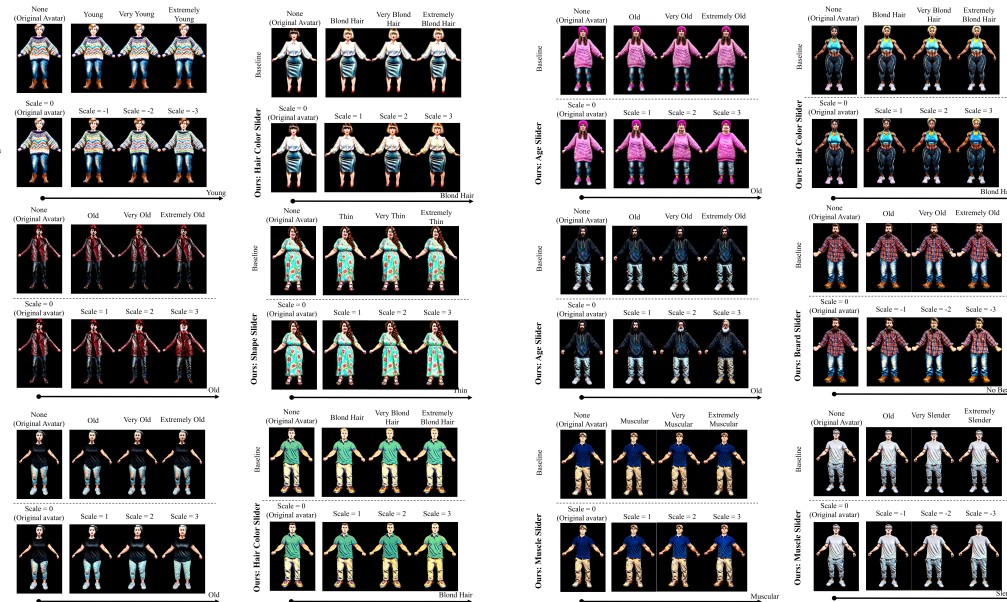

Figure 2: More qualitative comparisons with text-based editing baseline Liu et al. (2023)

## 2 MORE EXPERIMENTS

### 2.1 MORE QUALITATIVE COMPARISONS

In Fig. 4(a) of the main paper, we presented some comparisons of our method against text-based editing baselines Liu et al. (2023); Cao et al. (2023); Liao et al. (2024). Here, we show more results of baseline comparisons in Fig. 1 and Fig. 2. Note that these two figures show the same settings, but they are split into two figures due to page formatting issues. As observed in the results, our method offers precise controllability, enabling users to edit avatars according to their desired degree of the target concept. Our edited outcomes also tend to *better align with the provided descriptions*. On the other hand, we observe that the text-based baseline often struggles to make minor edits to the avatar, where it is difficult to edit the avatar differently according to changes in the degree of the input prompts.

### 2.2 MORE QUALITATIVE RESULTS

We have showed results of ACS across various concepts and avatars in Fig. 3(a) of the main paper. Here, we display even more results in Fig. 3. We observe that our method can effectively edit the source avatars precisely, while maintaining a good quality, as well as maintaining the identity of the avatars. We remark that we only fine-tune our LoRA slider within the range [-1,1], yet our ACS demonstrates strong extrapolation ability beyond this range, indicating a well-generalized understanding of the concept.

**Precise Editing Results.** In Fig. 3(b) of the main paper, we evaluated our method's ability to precisely edit the concepts. We show more results in Fig. 4, where we adjust the slider in smaller step sizes of 0.5. The results demonstrates our slider's ability at editing concepts precisely, over minor increments. We also remark that performing such minor and precise edits has been very challenging for previous text-based methods, which highlights our method's contributions for improving the controllability of editing 3D avatars.

**Multi-view Visualization of Edited 3D Avatars.** Next, in Fig. 5, we evaluate the quality of our edited 3D avatars by visualizing them from multiple views. The results show that our ACS is capable of maintaining the quality of the overall 3D avatar during editing. Furthermore, we also note that changes in concepts are often observable from other views as well, e.g., when editing the avatar to be thinner, the avatar's tummy is successfully flattened. Also, when we remove or shorten the beard,

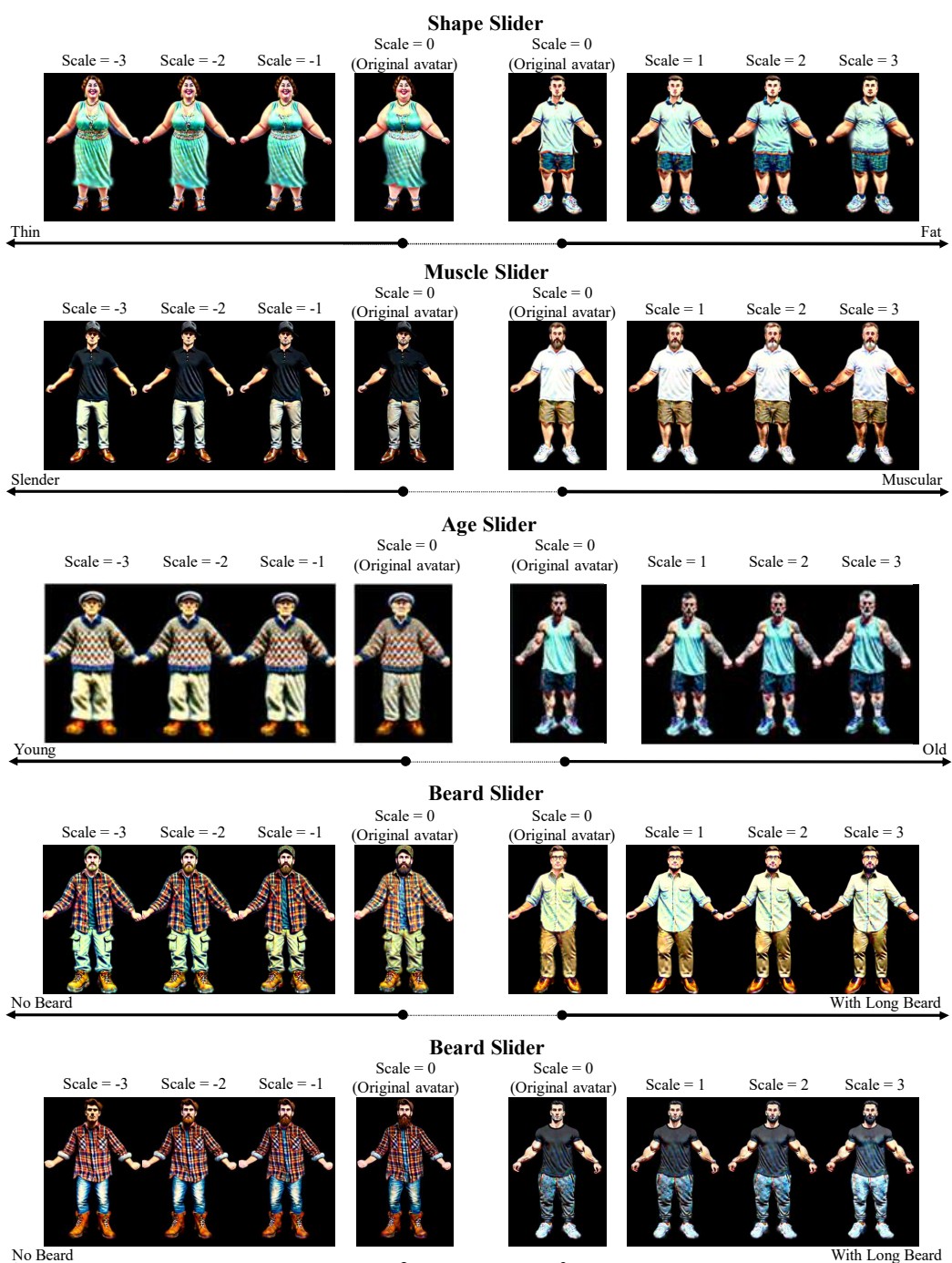

Figure 3: More editing results using various concept sliders.

we can also see from the side views that the beard is shortened and reduced in size as well. Overall, this further affirms the quality of avatars edited through our proposed method.

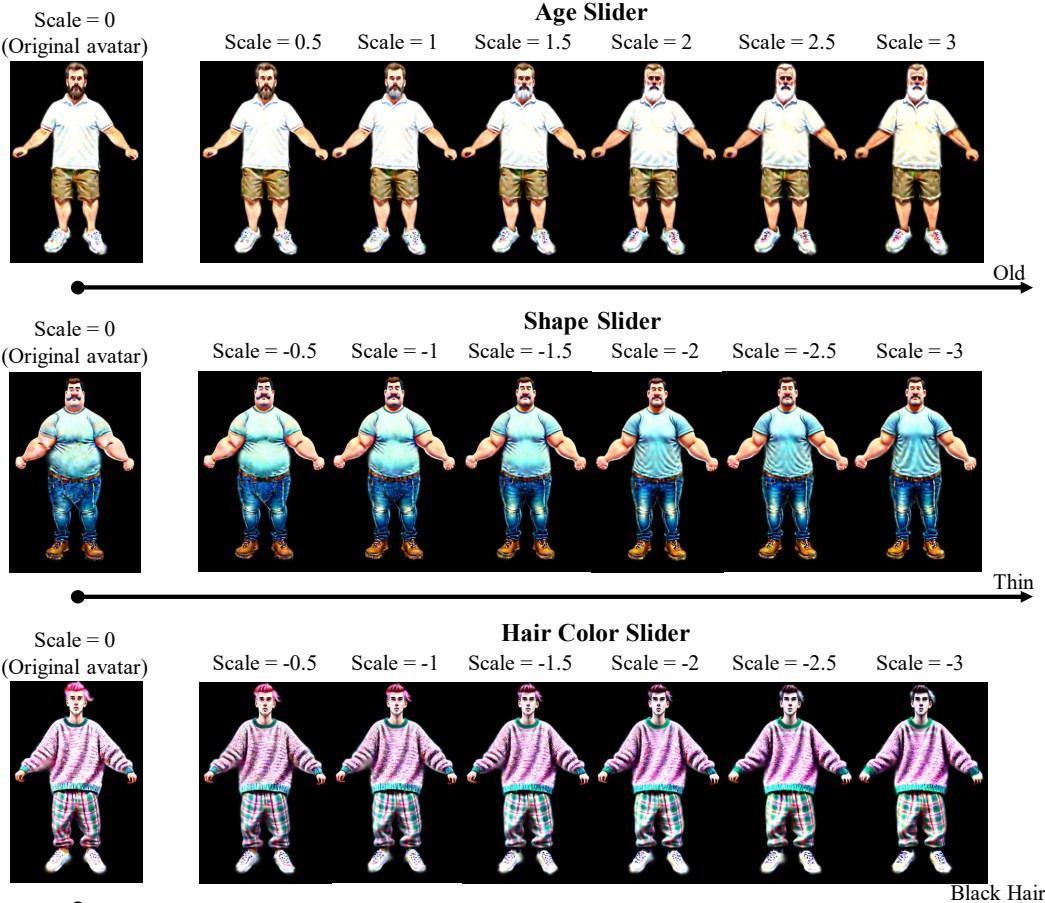

Figure 4: More qualitative evaluations of the precise controllability provided by our concept slider across several concepts. For each concept, we generated avatars across smaller steps (i.e., in steps of 0.5) along the concept sliders.

## 2.3 MORE RESULTS WITH COMPLEX CONCEPTS

In the main paper, as well as other sections of this Supplementary, we evaluate using simple concepts (e.g., hair length, body shape) as they are less subjective and thus easier to evaluate visually. However, we are also able to achieve good editing results with more complex and abstract concepts, as shown in Fig. 7. In Fig. 7, we edit the avatars using the concepts using the abstract concepts "cartoon-style" (top) and "sculpture-like" (bot). The results show that our method is also able to learn and precisely edit abstract concepts, which can be otherwise difficult to concretely describe. We remark that this is quite challenging since it requires a fine-grained control over the abstract concept, and our method is able to achieve this well.

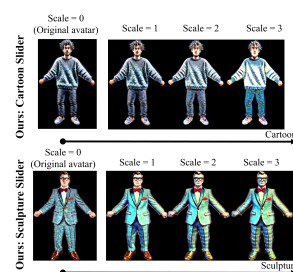

Figure 7: Visualization of results for more complex and abstract concepts.

## 2.4 MORE COMPARISONS WITH TEXT-BASED EDITING BASELINES

In Fig. 4(a) of the main paper, we showed results comparing our method against text-based editing baselines Liu et al. (2023); Cao et al. (2023); Liao et al. (2024). In that figure in the main paper, source avatars for each of the methods are generated according to their own generation pipeline, such that the avatars are in the correct representation for the editing. For example, the base avatar for TADA Liao et al. (2024) was originally generated with its text-based generation pipeline, while our method edits a 3DGS avatar generated using Liu et al. (2023); the text prompts used for the generation for the source avatar are the same. Here, for closer comparison of the editing on similar

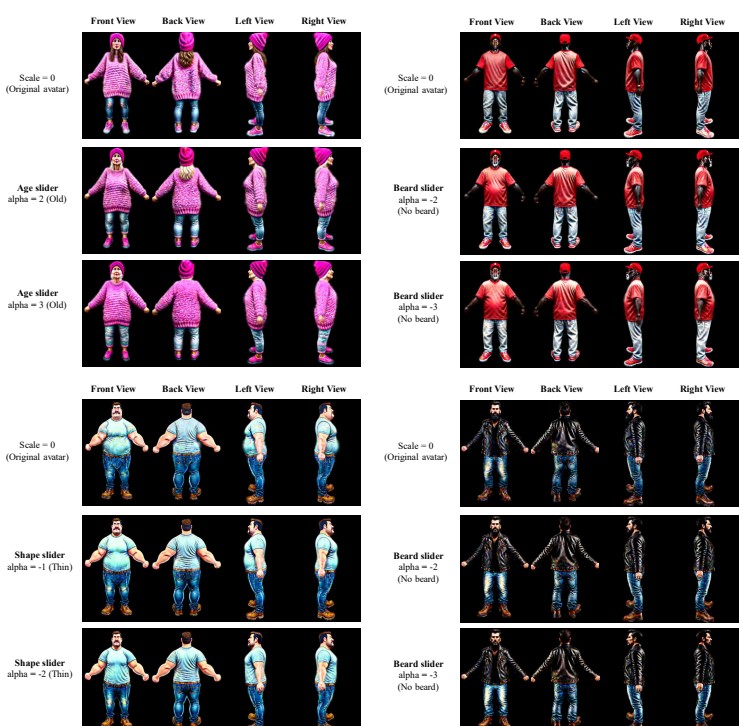

Figure 5: Multi-view visualization of edited 3D avatars.

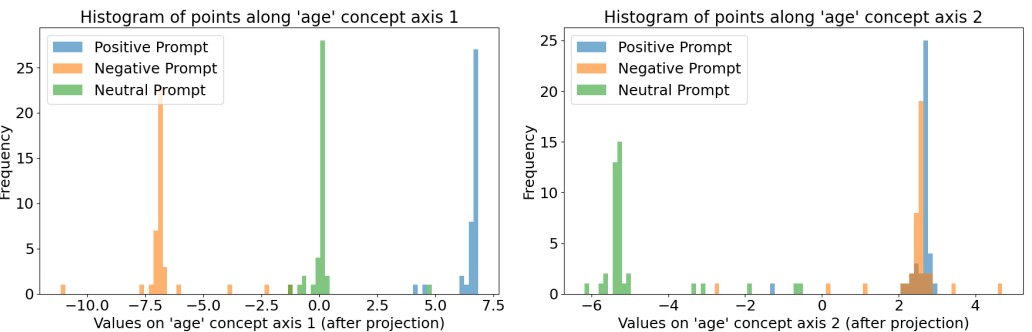

(a) Plot of values on the concept axes found by LDA (Age). Left is the main concept axis, and the second concept axis is on the right.

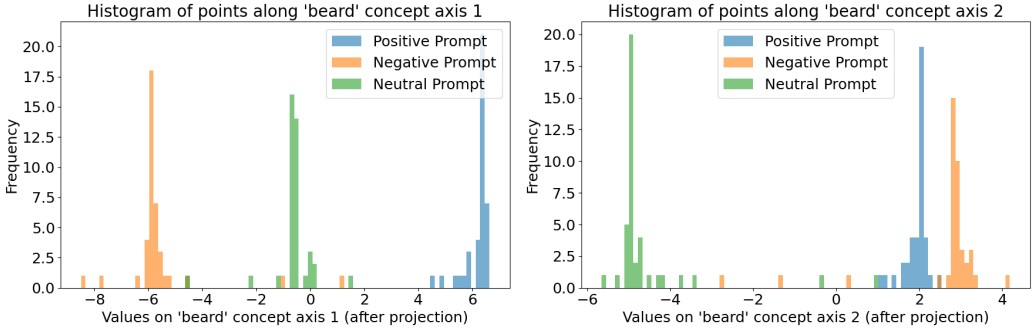

(b) Plot of values on the concept axes found by LDA (Beard). Left is the main concept axis, and the second concept axis is on the right.

Figure 6: Comparison of values on the concept axes found by LDA for different attributes.

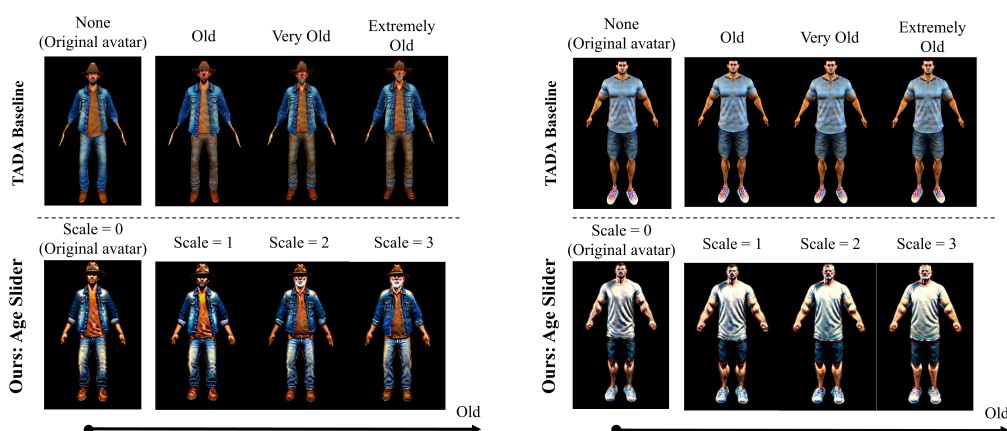

Figure 8: More editing results using base avatars generated by TADA Liao et al. (2024).

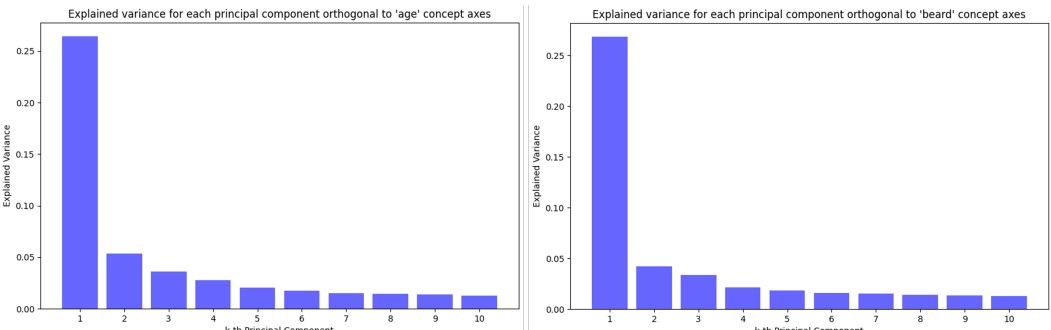

Figure 9: Plot of the variance explained by each attribute basis, e.g., principal component found by PCA. The plot on the left is for the 'age' concept, and the plot on the right is for the 'beard' concept. We observe that, the first few principal components are very important, but after the fifth principal component, the remaining principal components only face slight decreases. This suggests that using $K = 5$ is a suitable value.

source avatars, we further export the base avatar from TADA Liao et al. (2024) and convert to 3DGS representation, and run text-based SDS for a few iterations with our diffusion model. Results are shown in Fig. 8. We observe that our slider-based editing method yields avatars that adhere to concepts and preserves identity better. Additionally, we also yield higher quality avatars while being significantly faster (see Tab. 2 in main paper).

## 2.5 Qualitative Analysis of Concept Axes

In Sec. 4.1 of the main paper, we discussed how we leveraged LDA to identify the concept axes that were the most discriminative in separating the positive and negative classes. Here, in Fig 6, we analyze the concept axes for two of our sliders. Specifically, we sampled some latent features using positive, negative, and neutral prompts, which were not used in the original LDA fitting. Then, we projected those sampled latent features onto our concept axes found by LDA. As expected, we observe that, on the main concept axis (left), the positive features and the negative features are on opposite ends, with the neutral features roughly in the middle of the axis. According to the variance explained metric of LDA, the main concept axis for both of these sliders account for approximately 70% of the variance, which means it is a highly discriminative axis in the high-dimensional space. The second concept axis (right) found by LDA tends to find a different pattern to capture, and so the relative positions of the positive, neutral, and negative features are different from the main concept axis. Note that, since we design our Concept Sliding Loss to enforce the relative movements along the slider instead of actual locations of the features, our slider loss will encourage features to "slide" from one end to the other end by going through the "neutral" area, which applies even to the second concept axis. For instance, to slide a "positive" avatar to the "negative" side, our loss steers it first to the "neutral" area on the concept axis, before moving to the "negative side".

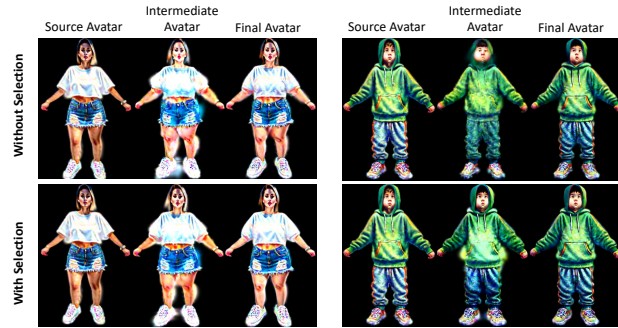

Figure 10: Ablation of Primitive Selection, using the "shape" slider. By using our Primitive Selection mechanism, we can increase efficiency significantly, without compromising on quality.

## 2.6 QUALITATIVE ANALYSIS OF ATTRIBUTE BASES.

In Sec. 4.2 of the main paper, we discussed how we computed the principal components of the features that are orthogonal to the found concept axes (which we call attribute bases), and here we analyze them further. In Fig. 9, we plot the variance explained of the first 10 principal components found by PCA, for the 'age' and 'beard' sliders. We observe that the first few principal components tend to be quite important, but after the fifth principal component, the variance explained values tend to stay constant. This suggests that, after the first five attribute bases, the rest of the principal components tend to have lower importance. Thus, we set $K = 5$ in our experiments.

Moreover, because the variance explained metric is a good indicator of the importance of an attribute, we also empirically found that it was helpful to additionally assign weights for the loss corresponding to each attribute basis based on the computed variance explained values. So we applied this weighting scheme for all the sliders we trained.

## 2.7 QUANTITATIVE ANALYSIS

We quantitatively evaluate our method with the a large vision-language model GPT-4o. Specifically, we applied our ACS to obtain 20 sets of edited avatars (4-7 avatars per set, 4 sets per concept), and randomize their sequence within

Table 1: Quantitative comparison against text-based editing baseline.

| Setting | Accuracy of VLM model ↑ |
|---|---|
| HumanGaussian Liu et al. (2023) | 10 % |
| Ours | 70 % |

each set. The sets with 7 avatars were obtained with the slider values (-3, -2, -1, 0, 1, 2, 3), and the sets with 4 avatars were obtained with the slider values (0, 1, 2, 3). Then, we sent in the sets of avatars into the GPT-4o model and asked it to rank the avatars according to the given concept. We also did the same for the same source avatars, but edited with a text-based baseline Liu et al. (2023), where the concepts are adjusted to varying degrees via adverbs (e.g., 'very', 'extremely'). Results are shown in Tab. 1. We observe that the VLM ranked our avatars much more accurately vs the text-based baseline. This shows that our method is able to edit avatars with good quality and according to the desired user-input concept. Furthermore, these results also show that, the edited avatars are still easily distinguishable with respect to the concept. Note that, *making precise and minor*, *yet distinguishable edits* is an extremely challenging task, and our method is able to make significant improvements in this direction, which highlights our contribution.

## 2.8 ABLATION OF PRIMITIVE SELECTION

In Sec. 4.3 of the main paper, we introduce a concept-sensitive primitive selection mechanism to reduce redundancy and enhance editing efficiency by selecting a small fraction of primitives. We visualize the edited avatar with and without this selection mechanism at their median and final editing steps, where the avatars edited with our method show less artifacts at areas unrelated to the target concept. The results, illustrated in Fig. 10, demonstrate that our selection mechanism accelerates the convergence process significantly, without negatively affecting the final quality. Notably, due to our

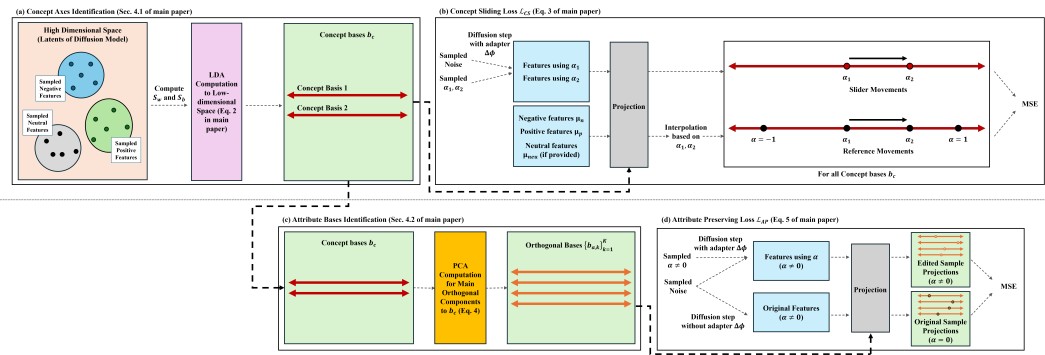

Figure 13: A detailed visual summary of our technical contributions in Sec. 4.1 and Sec 4.2 of the main paper.

focus on concept-sensitive Gaussians, our editing process with the selection mechanism converges quicker, in about 1200 steps, whereas the method without it takes approximately 3000 steps.

## 2.9 COMPARISONS AGAINST IMAGE-BASED EDITING METHOD

Here, we run comparisons against an image-based editing method Gandikota et al. (2024) which can perform attribute control in the 2D space. Firstly, we show comparisons in the image space, comparing the outputs using the LoRA adapter after the fine-tuning stage. For the 2D baseline Gandikota et al. (2024), we use the 'age' prompt in their demo, and apply it on the same diffusion model as us. For our method, we use the same "age" slider that is used for all visualizations in our paper. The results for image-based edits are in Fig. 11, which shows that our fine-tuned LoRA outperforms the baseline in terms of identifying the concept and maintaining the attributes of the original image. This is because of our proposed Concept Sliding Loss and Attribute Preservation Loss, which helps us to identify concepts better and preserve the identity better.

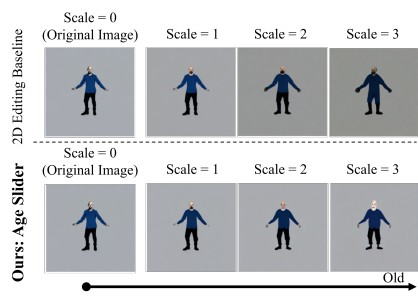

Figure 11: Comparison of our method against image-based editing method.

Moreover, we further apply the LoRA adapter trained by the 2D editing baseline into our SDS optimization pipeline, to optimize a 3D avatar. The comparison vs our method is shown in Fig. 12. We observe that the baseline 2D editing method does not capture the "age" concept as well as us, and also does not preserve various aspects of the original avatar well, e.g., face color. On the other hand, our method captures the age concept significantly more accurately and also preserves the identifying information (e.g., shirt color) much better.

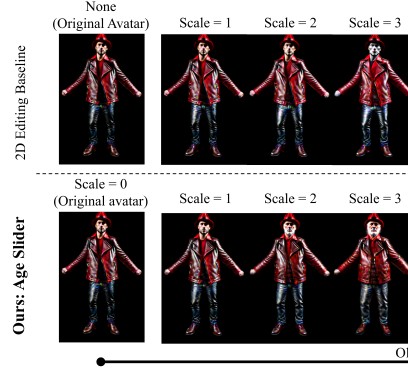

Figure 12: Comparison of our method against image-based editing method, when applied to 3D avatars.

## 3 MORE EXPLANATIONS OF OUR METHOD

To further improve the clarity of our proposed method, here we include figures visualizing our technical contributions in Sec. 4.1 and Sec. 4.2 of the main paper (Fig. 13).

In summary, as shown in Fig. 13(a), in order to fine-tune the adapter to have sliding capabilities, we first adopt LDA to compute the concept bases $b_c$. Then, as shown in Fig. 13(b), we randomly sample noise, as well as "slider values" $\alpha_1$ and $\alpha_2$, and run a diffusion step with our adapter $\Delta\phi$ using both $\alpha_1$ and $\alpha_2$. At the same time, the reference movements are also computed based on $\boldsymbol{\mu}_n$, $\boldsymbol{\mu}_p$ and

also $\boldsymbol{\mu}_{neu}$ if it is provided. These computed features are projected onto the concept bases, and our Concept Sliding Loss $\mathcal{L}_{CS}$ is the MSE between our model's movements from $\alpha_1$ to $\alpha_2$, and these reference movements. This facilitates the learning of the sliding capability, and movement along the slider.

As shown in Fig. 13(c), to improve the retaining of identifying attributes, we adopt a PCA-based approach to compute orthogonal bases $\{b_{a,k}\}_{k=1}^{K}$. Then, when using our adapter (i.e., slider value $\alpha \neq 0$) in a diffusion step, we also run the same diffusion step without the adapter where $\alpha = 0$ (Fig. 13(d)). Both of these denoised features are then projected onto the orthogonal bases, and our Attribute Preserving Loss $\mathcal{L}_{AP}$ is the MSE between the projected values on these bases. This helps to maintain key attributes of the avatar which are not related to the desired editing concept.

## 4    More Implementation Details

At the fine-tuning stage, when given a concept via text prompts $c_p$, $c_n$, and $c_{neu}$ we start by randomly generating $N_s = 40$ features for each concept using LoRA-unwrapped diffusion model $\boldsymbol{\epsilon}_{\phi_0}$ based on DDIM scheduler of $T_{DDIM} = 50$ time steps. We randomize the number of time steps to sample features, and keep the intermediate features from the latent features as $\{\mathbf{F}_{p,i,x}\}_{i=1,x=1}^{N_s,C}$, $\{\mathbf{F}_{n,i,x}\}_{i=1,x=1}^{N_s,C}$ and $\{\mathbf{F}_{neu,i,x}\}_{i=1,x=1}^{N_s,C}$, which are subsequently processed with LDA and iterative PCA analysis procedure to obtain the concept axes $\mathbf{b}_c$ and attribute bases $\{\mathbf{b}_{a,k}\}_{k=1}^{K}$. Meanwhile, we compute the mean vectors $\boldsymbol{\mu}_n$, $\boldsymbol{\mu}_p$ and $\boldsymbol{\mu}_{neu}$ for each set.

Regarding our positive, negative and neutral text prompts ($c_p$, $c_n$, and $c_{neu}$), they can be generated conveniently with a user-input pair of descriptions. For example, to obtain a "body shape" slider, the user only needs to input two ends of the concept, e.g., "fat" and "slim". In this case, the neutral prompt $c_{neu}$ can be "a human", while the positive prompt $c_p$ is "a fat human", and the negative prompt $c_n$ is "a slim human". However, in practice, we found better results when we slightly augmented these prompts according to gender. Specifically, we process the input descriptions into two sets of prompts ($c_{neu}$ ="a man", $c_p$ ="a fat man", $c_n$ ="a slim man") and ($c_{neu}$ ="a woman", $c_p$ ="a fat woman", $c_n$ ="a slim woman"). Then, we generate an equal number of latents for each gender, i.e., 20 sets of features for each gender. We note that such augmentation and processing of prompts can automatically and conveniently be done based on the input pair of descriptions.

We fine-tune our LoRA adapter while keeping the diffusion model fixed. Specifically, we input the positive, neutral, and negative prompts into the model to train the LoRA adapter over random diffusion timesteps. We freeze the parameters of original diffusion model and train the parameters of adapter for 1000 steps based on the $\mathcal{L}_{CS}$ and $\mathcal{L}_{AP}$ losses, as well as a quality preservation loss which matches the positive prompt output with the +1 value on the slider. Furthermore, since we are using positive, negative and neutral prompts the concept sliding loss $\mathcal{L}_{CS}$ in Eq. 3 of the main paper is adapted accordingly, where it becomes a piecewise loss function. Specifically, we use positive and neutral latents as training signals for movements between +1 and 0 respectively. Similarly, we use negative and neutral latents as training signals for the movements between -1 and 0 respectively. Moreover, the dimensions of $\mathbf{b}_c$ are also expanded, and the loss equation is also expanded to include the new features and concept means (e.g., neutral features).