# OpenReview forum: "Avatar Concept Slider: Controllable Editing of Concepts in 3D Human Avatars"
_ICLR.cc/2026/Conference — ICLR 2026 Conference Withdrawn Submission_

### Official Review · Reviewer_DeGG · 2025-10-16

**Soundness:** 3
**Presentation:** 2
**Contribution:** 2
**Rating:** 4
**Confidence:** 4

**Summary:**

This paper investigates text-based 3D avatar editing and introduces an avatar concept slider (ACS) method allowing precise control the desired degree of expression of a given concept. Specifically, a concept sliding loss is introduced, which leverage linear discriminant analysis (LDA) to pinpoint the concept-specific axes. An attribute preserving loss is introduced to extract the key attribute information orthogonal to the target concept using principal component analysis (PCA). In addition, based on the concept-sensitivity, a 3DGS primitive selection mechanism is introduced to improve efficiency.

**Strengths:**

+ Using a concept slider for controllable text-based 3D editing is practical.
+ The overall paper is technically sound and easy to understand.
+ Extensive editing results are provided to demonstrate the effectiveness and generalization ability of the proposed method.
+ The authors provide video demos to show the promising editing capabilities with good 3D consistency and controllable attribute changes.

**Weaknesses:**

- The proposed concept slider is somewhat complicated. Each new concept requires training a new LoRA adapter and manually designing a positive-negative text pair. The quality may rely on the manually defined opposite descriptions. It would be better to add a discussion or analysis on this.
- The experiments only showcase some simple concepts (simple attributes modification described using short phrases such as age, hair color/length and body shape). I am wondering if it is possible to adjust a combination of multiple concepts or more complex concepts using a single concept slider?
In addition, the attribute preserving approach shows low quality (e.g., woman’s jeans in Figures 4a and man’s jeans in Figure4b.
- Although the proposed concept-sensitive primitive selection shows promising efficiency (12 minutes as reported in Table 2), it heavily relies on fine-tuning the concept-specific adapter (~1 hour as mentioned in Section 5). Reporting only 12 minutes of editing time is misleading and somewhat unfair.
- The organization and writing could be further improved. Several paragraphs (particular Section 4) are somewhat wordy and redundant. It is also weird the implementation details become an independent section (i.e., Section 5). The text in tables and figures is extremely tiny, which seriously affects readability.
- There is no video demo to showcase the superiority compared to state-of-the-art methods.

**Questions:**

Please refer to Weaknesses.

---

### Official Review · Reviewer_kyf1 · 2025-10-28

**Soundness:** 1
**Presentation:** 1
**Contribution:** 2
**Rating:** 0
**Confidence:** 4

**Summary:**

The paper proposes avatar concept silder,  which aims to edit the 3D human avatars at a specified intermediate point between two extremes of concepts.The ACS can be represented in two stages. The first stage will finetuen a lora with a diffusion model to handle different alpha weights to combine positive and negative features. In the second stages, the 3D human avatars will be edited by the lora with given alpha. The results show the proposed method can modify the 3D avatar with some simple prompts.

**Strengths:**

1. The proposed method shows a better visualization compared with the baselines.

**Weaknesses:**

1. The writing is EXTREMELY HARD to follow.
2. The principle mentioned in the paper should be more general techniques for editing, like scene editing. It is hard to understand why the author only uses it in 3D human avatars, since LDA is irrelevant to "3D human avatars".
3. All the experiments only show two simple prompts, like fitness.
4. The details of training lora are missing. Part a of fig 2 is missing leading because it looks like the diffusion block only takes prompts as input.
5. The comparison with dramavatar is unfair. The author should provide the results with the same avatar.

**Questions:**

The usage of /cite in the paper looks like not correct.

Based on the overall writing quality of the paper, and considering the idea of sds is a little bit out-of-date, I tend to give a negative recommendation.

---

### Official Review · Reviewer_2GTW · 2025-10-30

**Soundness:** 3
**Presentation:** 3
**Contribution:** 3
**Rating:** 6
**Confidence:** 3

**Summary:**

This paper proposes a 3D avatar editing method: avatar concept slider (ACS), which allows editing of semantic concepts in human avatars towards a specified intermediate point between two extremes of concepts, akin to moving a knob along a slider track. The authors introduce a Concept Sliding Loss, an Attribute Preserving Loss and 3D Gaussian Splatting primitive selection mechanism to try achieving editing with the concept slider.

**Strengths:**

Precisely editing a human avatar at a specified intermediate point between two extreme concepts is an interesting and practically valuable topic.

The method proposed by the authors is simple and appears to have high convergence efficiency.

**Weaknesses:**

From the examples provided by the authors, the concepts seem quite limited. How many concepts can your method support? Are the concepts limited to a predefined set, or can they be arbitrary open-vocabulary terms? If the concepts are indeed limited, how can the work be justifiably termed “full-body avatar editing”? This is a critical point that affects the practical significance and generalizability of the method.

The authors claim to address the challenging problem of isolating and editing only the desired concept without altering other identifying information of the avatar. They introduce an “Attribute Preserving Loss” that attempts to achieve this through PCA and feature orthogonalization. However, the proposed method does not seem to guarantee that different attributes are truly decoupled. Could the authors provide more rigorous proof or a more comprehensive experimental analysis to demonstrate the effectiveness of their decoupling approach?

Regarding the User Study, were the 16 sets of edited avatars and their corresponding descriptions prepared in advance by the authors? If so, this raises a concern about potential bias. How did you ensure that the selection of examples and descriptions does not favor your method? To mitigate this, could the editing text prompts be provided by the test participants themselves, rather than being pre-selected by the authors?

**Questions:**

The author needs to respond to the Weakness.

---

### Author Response · Authors · 2025-11-14

We sincerely thank all reviewers and AC for your time in reviewing our submission. We respectfully believe our paper merits a more balanced evaluation, specifically we feel that the comments provided by reviewer kyf1 did not fully support the score assigned. Thus, we are withdrawing our submission.

---

### Note · Authors · 2025-11-15

I have read and agree with the venue's withdrawal policy on behalf of myself and my co-authors.